# Transcutaneous Auricular Vagus Nerve Stimulation Facilitates Cortical Arousal and Alertness

**DOI:** 10.3390/ijerph20021402

**Published:** 2023-01-12

**Authors:** Yuxin Chen, Xuejing Lu, Li Hu

**Affiliations:** 1CAS Key Laboratory of Mental Health, Institute of Psychology, Chinese Academy of Sciences, Beijing 100101, China; 2Department of Psychology, University of Chinese Academy of Sciences, Beijing 100049, China

**Keywords:** transcutaneous auricular vagus nerve stimulation (taVNS), cortical arousal, alertness, attention, alpha oscillations, microstates

## Abstract

Transcutaneous auricular vagus nerve stimulation (taVNS) is a promising noninvasive technique with potential beneficial effects on human emotion and cognition, including cortical arousal and alertness. However, it remains unclear how taVNS could improve cortical arousal and alertness, which are crucial for consciousness and daily task performance. Here, we aimed to estimate the modulatory effect of taVNS on cortical arousal and alertness and to reveal its underlying neural mechanisms. Sixty subjects were recruited and randomly assigned to either the taVNS group (receiving taVNS for 20 min) or the control group (receiving taVNS for 30 s). The effects of taVNS were evaluated behaviorally using a cue-target pattern task, and neurologically using a resting-state electroencephalogram (EEG). We found that taVNS facilitated the reaction time for the targets requiring right-hand responses and attenuated high-frequency alpha oscillations under the close-eye resting state. Importantly, taVNS-modulated alpha oscillations were positively correlated with the facilitated target detection performance, i.e., reduced reaction time. Furthermore, microstate analysis of the resting-state EEG when the eyes were closed illustrated that taVNS reduced the mean duration of microstate C, which has been proven to be associated with alertness. Altogether, this work provided novel evidence suggesting that taVNS could be an enhancer of both cortical arousal and alertness.

## 1. Introduction

*Cortical arousal* refers to the non-specific activation of the cerebral cortex associated with sleep-wake states [1]. It is controlled by the neuromodulatory system of the brainstem nucleus [2], to which the afferent branch of the vagus nerve projects [3]. With a similar neural basis, *cortical arousal* is sometimes confused with *alertness* [1,4]. From a psychological view, the alerting system is an essential part of attention and is in charge of maintaining optimal vigilance and performance [1,5]. Notably, there is no consensus among researchers in the search for an ideal term to describe states of cortical activation. Thus, we preferred to define the term used for the spontaneous activation level of the cerebral cortex as “cortical arousal”, and the ability to maintain performance and keep vigilance as “alertness” in the present study. Both cortical arousal and alertness play crucial roles in our daily life. Exploring the modulatory effect of vagus nerve stimulation (VNS) on these two states may provide a promising neuromodulation technique for improving drowsiness under heavy work in the healthy population. Furtherly, it will inspire therapies for treating patients with disorders of consciousness [6,7], as well as attention deficit and hyperactivity disorder (ADHD, [8]).

It has been reported that VNS could lead to an enhancement of *arousal* [9,10], as reflected by an increased activation of locus coeruleus-norepinephrine (LC-NE, [11]), a large pupil dilation [12], and a reduction of alpha (α) oscillations in an EEG [12,13]. As an electrophysiological indicator of cortical arousal [14,15,16], α oscillations were demonstrated to be attenuated after 3.4 s transcutaneous auricular vagus nerve stimulation (taVNS) [12] and after two 120 s transcutaneous cervical vagus nerve stimulations (tcVNS) [13]. However, there was a lack of investigation on the modulatory effect of a long-bout of continuous transcutaneous VNS on cortical arousal under a resting state.

In addition, transcutaneous VNS was also assumed to have an enhancement effect on the alerting system, which was frequently reflected by the performance of detection tasks (e.g., reaction time, RT [17,18]). Recently, Lerman et al. [17] revealed that 120 s of tcVNS could reduce the RT for target detection. Nevertheless, it was difficult for tcVNS to selectively stimulate the cervical vagus nerve via the skin of the neck, particularly the afferent nerve [19]. In contrast, the taVNS could, nearly, only activate the afferent nerve of the auricular branch of the vagus nerve (ABVN) through the cymba conchae [3,19]. To our knowledge, studies that explicitly reported an enhancement effect of taVNS on alertness level were limited to patients [18,20]. However, it was hard for patient studies to set a perfect control manipulation and isolate the pure effect of taVNS from other effects of standard therapies. Therefore, it is also necessary to assess the modulatory effect of taVNS on alertness in a healthy population.

To this end, the present study aimed to test the modulatory effects of left taVNS on alertness by the general performance of a cue-target task, and cortical arousal by spontaneous α oscillations under the resting state. Additionally, the global brain dynamics under the resting state, before and after taVNS, were estimated through the microstate analysis, which provided topographical alternations of networks (e.g., the cingulo-opercular system) associated with maintaining cortical arousal and alertness [21,22,23]. We hypothesized that both cortical arousal and alertness would be enhanced after taVNS. Notably, the left taVNS could also increase GABA-A activity in the contralateral (i.e., right) sensorimotor cortex [24], which would enhance the ipsilateral (i.e., left) hand’s automatic motor inhibition [25]. Jointly considering the influence of both the activation of the LC system and GABA-A activity, we therefore hypothesized that the left taVNS could enhance cortical arousal in the left brain regions and accelerate the right-hand reaction time.

## 2. Materials and Methods

### 2.1. Subjects

Sixty healthy right-handed subjects, who were unacquainted with noninvasive neuromodulation, were recruited and randomly assigned to the taVNS group (13 males, 17 females) or the control group (13 males, 17 females). One subject was excluded from the taVNS group because of incomplete data collection during the EEG recording. The two groups were matched in terms of age, sex ratios, and years of education (see Appendix A). All subjects declared no history of physical or psychiatric/neurologic diseases. They were asked to abstain from alcohol, aspirin, ibuprofen, and other analgesic drugs for at least 24 h prior to the experiment. Written informed consent was obtained from each subject prior to the experiment. The experimental procedures were in accordance with the ethical standards of the local ethics committee.

### 2.2. The Cue-Target Pattern Task

Subjects were asked to perform a cue-target pattern task (i.e., a dot-probe task) in both the pre-test and post-test phases (see Figure 1a,b). Briefly, a solid white dot as the target was presented on the screen’s left or right side, followed by cues containing horizontally distributed face pairs with different valences (e.g., neutral, sad, or happy). Subjects were required to indicate whether the dot appeared on the left or the right side as quickly and accurately as possible by pressing the “d” or “j” button with the left or right index finger, respectively. The accuracy and RT for each trial were recorded. Trials with incorrect responses or an unreasonable short or long RT (<100 ms or >1500 ms) were excluded from further analysis. To examine the laterality effect of taVNS on rapid changes in alertness under the cue-target paradigm, the averaged RT of the left-hand or right-hand responses to targets were calculated separately for each subject and test phase.

### 2.3. taVNS and Control Intervention

In line with previous studies [11,12,26], stimulating electrodes were placed at the left cymba conchae and concha that the ABVN mainly innervated ([3,27,28]; see Figure 1c). For the taVNS group, electrical stimulation was generated with a monophasic square wave (pulse width: 250 μs, pulse frequency: 25 Hz) by an electrical stimulator (SXC-4A, Sanxia, China) and applied to subjects for 20 min in the intervention phase. By contrast, for the control group, the stimulation with similar parameters only lasted for the first 30 s of the intervention phase. Stimulation intensity was individually customized via the “method of limits” procedure [29] to meet a strong but non-painful sensation (see Appendix A). The stimulus intensity was adjusted at the 7th min and 14th min after beginning the stimulation to avoid adaptation for the taVNS group. Subjects from the control group followed the same procedure to ensure they were unaware of receiving an inactive intervention. All subjects were required to keep their eyes open, sit comfortably, and avoid frequent movement during the intervention phase.

### 2.4. Experimental Procedure

After the EEG set-up (see below), the individually customized stimulus intensity was determined. To rule out the possible influence of emotional trait/state on taVNS effects, the differences in the depression level, as well as anxiety level between the two groups, was measured using the Chinese version of the Beck Depression Inventory-II (BDI-II; [30]) and the Trait-Anxiety Inventory subscale (T-AI) from the State-Trait Anxiety Inventory [31], respectively. Subsequently, a pre-test phase, a 20 min intervention phase, and a post-test phase were successively administered (see Figure 1a). Each test phase included a dot-probe task, a resting state EEG recording (a three-minute open-eye resting state and a three-minute close-eye resting state), and a Chinese version of the Positive and Negative Affect Scale (PANAS; [32]) assessment.

### 2.5. EEG Recording

EEG data were continuously recorded through all phases via 32 Ag-AgCl scalp electrodes placed according to the International 10–20 system (ANT, Neuro, The Netherlands). The signals were recorded and online filtered at 0.3–30 Hz with reference to CPz at a sampling rate of 1000 Hz. All electrode impedances were below 10 kΩ.

### 2.6. EEG Data Pre-Processing and Analyses

EEG data were preprocessed and analyzed using the open-source toolbox EEGLAB [33] performed in the MATLAB environment (2014; MathWorks, Natick, MA, USA). Another notch filter was applied from 24 to 26 Hz to suppress the residual artifact stemming from the 25 Hz taVNS. In both the pre-test and post-test phases, the data from 30 to 180 s of both open- and close-eye resting states for each subject were selected for the following spectral and microstate analyses.

#### 2.6.1. Resting State EEG Power Spectrum Analysis

EEG epochs were extracted using a 2000 ms time window, while the epochs contaminated by gross artifacts (e.g., muscle artifacts) were removed. Eyeblinks and movements were also corrected using an independent component analysis (ICA) algorithm [33]. After that, the EEG data were re-referenced to the averaged signals of bilateral mastoids. EEG data for open- and close-eye resting states were separately processed.

Power spectral density (PSD) for each frequency point of each epoch over each electrode was estimated through the periodogram procedure and then averaged across the epochs to obtain single-subject level spectra in two phases, respectively. Afterward, the group-level PSD at each electrode was calculated by averaging the EEG spectra across subjects in the same group. Considering the close relationship between α oscillations and cortical arousal [14,15,16], we selected α band as the frequency of interest. Since there are two distinct alpha components (i.e., one is the low-α component peaking at ~9 Hz and maximal at the occipital cortex; another is the high-α component peaking at ~11 Hz and maximal at the somatomotor cortex [34,35]), we analyzed the modulatory effect of taVNS on the low- and high-α oscillations, separately.

#### 2.6.2. Microstate Analysis

For microstate analysis, a 2–20 Hz notch filter was used in previous studies [21]. Denoised EEG data were then re-referenced to the average reference. The standard deviation of potentials across electrodes, identified as Global Field Power (GFP), was calculated at each time point. Afterward, an atomize and agglomerate hierarchical clustering (AAHC) algorithm [36] was applied to cluster maps of GFP peaks at the subject level. We combined all subjects’ data across the groups and test phases to obtain a global template for the following back fitting. In line with previous studies [21,37], four classes of microstates were identified. The global template was back fitted to the EEG data for each subject by calculating the correlation between the topographic distribution of GFP peaks and each global microstate, obtaining the single subject sequence of microstate. Characteristics of microstates, such as occurrence rate, mean duration, time coverage for each microstate, and transition probability between the microstates were calculated at single subject level for further analyses.

### 2.7. Statistic Analyses

To demonstrate the modulatory effect of taVNS, independent sample t-tests were performed on the post−pre differences in RT, α-oscillation PSD, and characteristics of four microstates between the two groups. Cohen’s d was calculated to illustrate the effect size for the t-tests. For α-oscillation PSD, *p* values were adjusted by Bonferroni correction for multiple comparisons across electrodes. Furthermore, we adopted the partial correlation analysis to interpret the relationship between the effect of the taVNS on the task performance (i.e., RT) and on resting-state EEG signals (α-oscillation PSD and characteristics of microstates), controlling the influence of individual factors (e.g., sex and taVNS threshold) and emotional factors (i.e., the BDI-II score, the T-AI score, and the change of PANAS score).

## 3. Results

### 3.1. taVNS Effects on the Affective Score

No significant group difference was observed in BDI-II (taVNS: 8.620 ± 9.959 [mean ± SD, the same hereinafter]; control: 7.870 ± 8.059; t_(57)_ = 0.320, *p* = 0.750) and T-AI scores (taVNS: 40.590 ± 11.990; control: 39.230 ± 8.232; t_(57)_ = 0.507, *p* = 0.614), suggesting that the two groups were comparable in terms of depression and trait anxiety levels prior to the intervention phase. Furthermore, the modulatory effect of taVNS on affective states was not significant, as no group difference was found in the post–pre changes of positive (taVNS: −2.103 ± 3.792; control: −1.633 ± 3.792; t_(57)_ = −0.476, *p* = 0.636) or negative PANAS scores (taVNS: 1.207 ± 3.560; control: 0.833 ± 4.411; t_(57)_ = 0.357, *p* = 0.722).

### 3.2. taVNS Effects on the RT

The taVNS group showed a significantly larger decrease of RT than the control group for right-hand responses (taVNS group: −20.545 ± 20.844 ms; control group: −8.860 ± 20.124 ms; t_(57)_ = 2.191, *p* = 0.033, Cohen’s d = 0.571) but not for left-hand responses (taVNS group: −14.605 ± 18.191 ms; control group: −8.920 ± 17.819 ms; t_(57)_ = 1.213, *p* = 0.230; see Figure 1d).

### 3.3. taVNS Effects on the Resting-State α Oscillations

For the close-eye resting state, there was a widespread decrease in the PSD for high-α oscillations (i.e., ~11 Hz) after taVNS but not after the control intervention (see Figure 2a). After the multiple comparison correction, the PSD of high-α oscillations significantly reduced at two adjacent electrodes (i.e., C3 and CP5 electrodes; see Figure 2b). To further explore the relationship between the resting state high-α oscillations and task performance associated with active taVNS, the post–pre PSD difference at the C3 and CP5 electrodes was calculated for each subject (taVNS: −0.836 ± 2.022 dB; control: 0.706 ± 1.336 dB; t_(57)_ = 3.466, *p* = 0.001; see Figure 2c,d). In contrast, there was no significant difference between groups for post–pre PSD difference of low-α oscillations (i.e., ~9 Hz; all electrodes *p* > 0.05).

For the close–open resting state, neither the post–pre PSD difference of low-α oscillations nor that of high-α oscillations was significantly different between the two groups (all electrodes *p* > 0.05).

### 3.4. taVNS Effects on the Microstates

The main procedure of microstate analysis was showed in Figure 3a–d. Four microstates under the close-eye resting state for back fitting were displayed in Figure 3c,d. No group difference was found on the global explained variance (GEV), which reflects the percent variance explained by the above microstates, in either the pre-test (taVNS: 78.756 ± 4.955%; control: 78.535 ± 5.749%; *p* = 0.875) or post-test phase (taVNS: 79.379 ± 5.644%; control: 79.902 ± 5.903%; *p* = 0.729; see Appendix A). A significant difference between the taVNS group (2.145 ± 6.508 ms) and the control group (−1.105 ± 4.895 ms) was found on the mean duration difference of microstate C (t_(57)_ = 2.173, *p* = 0.034, Cohen’s d = 0.566; see Figure 3e). No other significant effect of the taVNS was found under close-eye resting state.

GEV was comparable between the two groups in both the pre-test (taVNS: 75.983 ± 3.675%; control: 75.455 ± 4.034%; *p* = 0.601) and post-test phases (taVNS: 77.198 ± 3.633%; control: 76.327 ± 4.244%; *p* = 0.401) under the open-eye resting state. However, differing from the close-eye resting state, null effect of taVNS on each microstate was found under the open-eye resting state (all *p* > 0.05).

### 3.5. Parietal Correlations between Behavioral Performance and EEG Features

We adopted the partial correlation coefficient to explore the relationship between the RT difference, the PSD difference of high-α oscillations, and the mean duration difference of the microstate C associated with taVNS under the close-eye resting state. There was a significant partial correlation between the RT difference and the PSD difference of high-α oscillations (r_(21)_ = 0.547; *p* = 0.007; Figure 4), after controlling the sex, taVNS threshold, BDI-II score, T-AI score, and PANAS score differences. However, we did not find a significant correlation between the RT difference and the mean duration difference of the microstate C (r_(21)_ = 0.355; *p* = 0.097).

## 4. Discussion

Resting-state EEG signals reflect spontaneous fluctuations of brain networks at rest [35], which are crucial for endogenous or top-down constraining of sensory-, cognitive- and motor-driven activities and mutual communications [36]. To explore the modulatory effect of taVNS on cortical arousal and alertness, the present study examined the changes in PSD of high-α oscillations (reflecting the thalamocortical communication) and microstates (capturing the global brain activity on the scalp that remains semi-stable) before and after interventions between the taVNS and the control groups. In addition to the behavioral results, where taVNS accelerated the responses to the cued targets, we found that taVNS attenuated spontaneous high-α oscillations at the C3 and CP5 electrodes and increased the mean duration of the microstate C, a microstate demonstrated to be associated with alertness. Importantly, a positive correlation was observed between the post–pre RT changes in response to the targets, and spontaneous high-α oscillation changes, suggesting a possible link between the taVNS-modulated effects behaviorally and neurologically.

One of the most salient findings in the present study was a widespread decrease of the PSD of high-α oscillations after taVNS but not control manipulation under the close-eye resting state. This decrease was dominantly at the sensorimotor cortex area (i.e., C3 and CP5 electrodes) in the left hemisphere (ipsilateral to the taVNS stimulated site; see Figure 2b). This finding was in line with previous studies showing a reduction in spontaneous α-band activities after a single bout of VNS [38,39] or transcutaneous VNS modulation [12,13,40]. As an indicator of the idling state [41], resting-state α oscillations were thought to be associated with the inhibitory effect on both the neural spike time and the firing rate, which determines cortex excitability [42,43,44]. Therefore, the taVNS-modulated decrease in spontaneous α oscillations in the present study might be associated with a wide range release of the inhibition and the higher arousal level of cortices, especially at the sensorimotor cortex, which could further increase the detection rates of exogenous stimuli and improve execution [43,44,45]. This finding is consistent with previous neuroimaging evidence showing a significant BOLD signal increase during and after taVNS around the pre- and post-central gyrus [11,46,47]. More importantly, the taVNS effect on resting-state high-α oscillations at the left sensorimotor cortex positively correlated with task performance improvement (see Figure 4). In other words, the greater high-α oscillation decrease after taVNS, the faster the response to the cued target compared to those in the pre-test phase. The facilitation in RT of the cued target represented an alertness improvement after taVNS, which has been discussed in detail previously [48]. Thus, this positive correlation revealed the extent of the taVNS modulatory effects on resting state cortical arousal around the sensorimotor cortex, and the alertness level to maintain task performance, were closely related. In this view, we inferred that taVNS might modulate cortical arousal and alertness through the same neural circuit (e.g., the LC-NE system [49]).

In addition to the reduced high-α oscillation PSD at the sensorimotor cortex, taVNS also induced some changes in microstates. In line with previous studies, we identified four archetypal microstates (i.e., microstates A–D), which were consistently reported in many previous studies [21,37,50]. Different microstates showed different scalp distributions and explained the global activity and coordination pattern of discrete brain states [21,37]. More importantly, there was a significantly increased mean duration of microstate C after taVNS modulation compared to the control manipulation (see Figure 3e). Microstate C was proposed to be associated with the saliency network, and the anterior default mode network, as well as the cingulo-opercular system, revealed by the correlation between the microstate C and the activity of the dorsal anterior cingulate cortex, the bilateral inferior frontal cortices, and the right insular [21,23]. The default mode network was assumed to be a task-negative network [21,51], but it was also confirmed to be responsible for monitoring the external environment, maintaining self-consciousness, and emotional processing under the awake and resting states [52,53]. Similarly, the central function of the cingulo-opercular system was maintaining alertness or vigilance [43,54], which was essential for task performance [5]. As a support, the duration of microstate C was reported to be positively linked to the vigilance level [55] and more time spent in maintaining alertness [56]. Thus, taVNS might facilitate alertness maintenance by increasing the duration of microstate C.

It should be noted that, compared to other microstates, microstate C was driven by stronger widespread α-band activities, especially at the occipital cortex [57]. Pascual-Marqui et al. [58] measured 203 subjects’ microstates under the eye-close resting state, and they found that microstate C was strongly associated with the predominant normalized generator distribution at the high-α oscillation (around 11 Hz). Additionally, α oscillations [43] and microstate C [23] marked the alertness level through the cingulo-opercular system. Besides, a previous study [59] revealed a positive association between the duration of microstate C and the occipital alpha power. However, we did not find a direct relationship between changes in high-α oscillations at the sensorimotor cortex and the mean duration of microstate C. Instead, we observed a decrease in the high-α oscillations but an increase in the mean duration of microstate C after taVNS, suggesting distinct neural mechanisms and activity patterns for sensorimotor α oscillations and occipital α oscillations [34,35,60]. It was reported that sensorimotor α oscillations have differentiable topographic distributions, neural generators, and functions with occipital α oscillations [60,61]. In addition, the disassociation of the taVNS modulatory effects on high-α oscillations and microstate C suggested that there could be at least two underlying neural modulation mechanisms. Indeed, taVNS could influence neuron firing or interactions in one or more systems involved in regulating the activity of α oscillations or the microstate C through different mechanisms [11,24,62,63], which need well-designed animal studies or neuroimaging studies to explore further.

Although we demonstrated a modulatory effect of taVNS on cortical arousal under the close-eye resting state, a null effect was found during the open-eye resting state. This modulation difference might be caused by the difference in the scope for improvement in cortical arousal and alertness between the two states, especially in the healthy population who were in a bright room under the task-free condition. As is known, α oscillation power is much higher when eyes are closed than when eyes are open [64]. Subjects could be staying awake and alert effortlessly and not easily falling into drowsiness under an eyes open state, resulting in a ceiling effect of taVNS modulation. Therefore, future work should focus on special populations (e.g., patients suffering from ADHD or disorders of consciousness) or specific application situations (e.g., long-term heavy work).

Since the taVNS was applied through the skin, which could also evoke cortical arousal changes, particularly around the sensorimotor cortex, it is possible that the taVNS effects observed in the present study resulted from the obvious difference in somatosensory stimulation between the taVNS and the control conditions. However, in our previous study, with a similar experimental design (the effect of the intervention was assessed by the comparison between pre-test and post-test), we observed that strong but non-painful transcutaneous electrical nerve stimulation could evoke strong body sensation but no significant changes in EEG oscillations [65]. In other words, even if there was an effect of strong body sensory stimulation, it cannot be sustained for minutes (there was a several-minute interval between the end of the intervention phase and the start of the close-eye state in the post-test phase). Therefore, we believe the enhancement of cortical arousal and alertness in the present study was mainly driven by vagus nerve activation rather than strong body sensory stimulation. Future work should be performed to provide direct evidence to support this statement.

## 5. Conclusions

In conclusion, the present study sheds new light on the modulatory effects of taVNS on cortical arousal and alertness, evidenced by the spontaneous local and global brain neural activities as well as improved attentional task performance. As a noninvasive modulatory approach, taVNS exhibits great potential in enhancing alertness-related cognitive functions in healthy populations.

## Figures and Tables

**Figure 1 ijerph-20-01402-f001:**
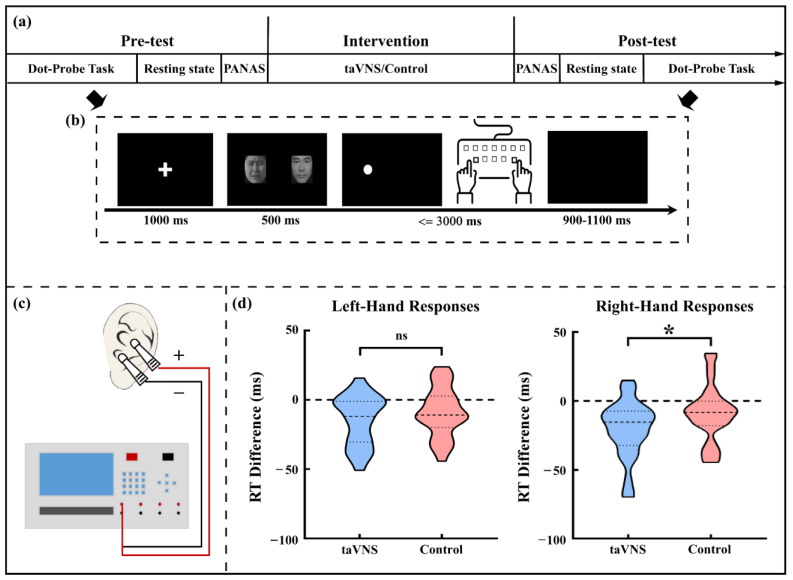
Experimental procedure and behavioral results. (**a**) Description of the experimental procedure; (**b**) Description of the single trial procedure in the dot-probe task; (**c**) Schematic of the taVNS stimulator and the stimulating sites; (**d**) The comparison of RT difference (post–pre) for left- and right-hand responses between the two groups. * *p* < 0.05; taVNS, transcutaneous auricular vagus nerve stimulation; RT, reaction time; ns, non-significant.

**Figure 2 ijerph-20-01402-f002:**
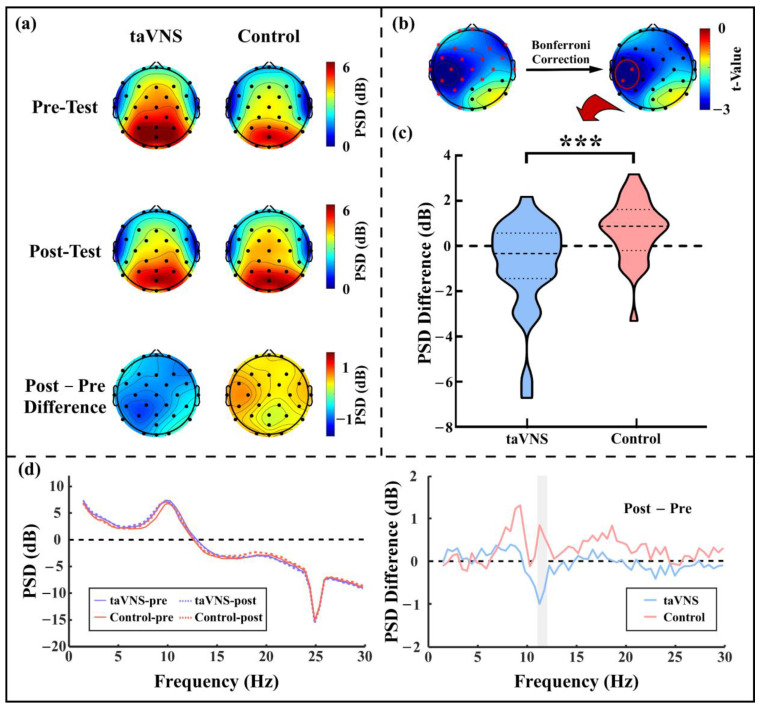
The high-α oscillation PSD under the close-eye resting state. (**a**) PSD topographies in the pre-test and post-test phases and the post–pre PSD difference topographies of high-α oscillations for the taVNS group and control group; (**b**) Topography of the t-value obtained from independent sample t-tests at each electrode for high-α oscillation PSD difference; (**c**) High-α oscillation PSD difference for the taVNS group and the control group; (**d**) Group average spectra and their post–pre differences at the C3 and CP5 electrodes for the taVNS group and the control group. Red asterisks in (**b**) represents significant differences between the taVNS group and the control group; the shadow region in (**d**, right panel) represents the frequency range of high-α oscillations; *** *p* ≤ 0.001.

**Figure 3 ijerph-20-01402-f003:**
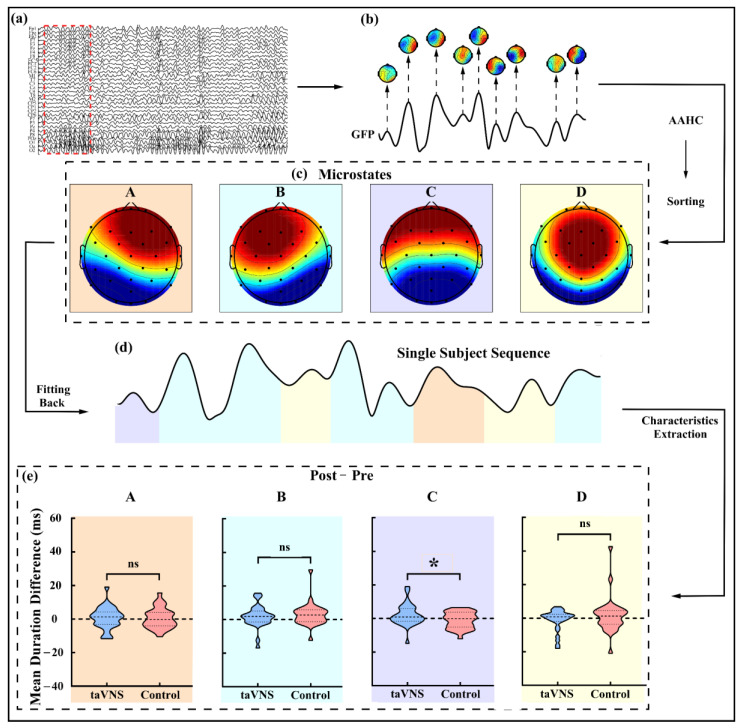
Microstate analysis procedure and mean duration difference of each microstate under the close-eye resting state. (**a**) EEG data after preprocessing; (**b**) GFP of signals marked by red dashed box in (a) and topographies at the latencies of GFP peaks; (**c**) Four microstates (i.e., microstates A–D) obtained by integrating all subjects’ EEG data after AAHC and sorting; (**d**) Single subject sequence obtained by back fitting from global microstate templates; (**e**) Mean duration difference for each microstate (i.e., microstates A–D, respectively, marked in different colors) under the close-eye resting state was extracted and calculated by single subject sequence. GFP, global field power; AAHC, atomize and agglomerate hierarchical clustering. * *p* < 0.05.

**Figure 4 ijerph-20-01402-f004:**
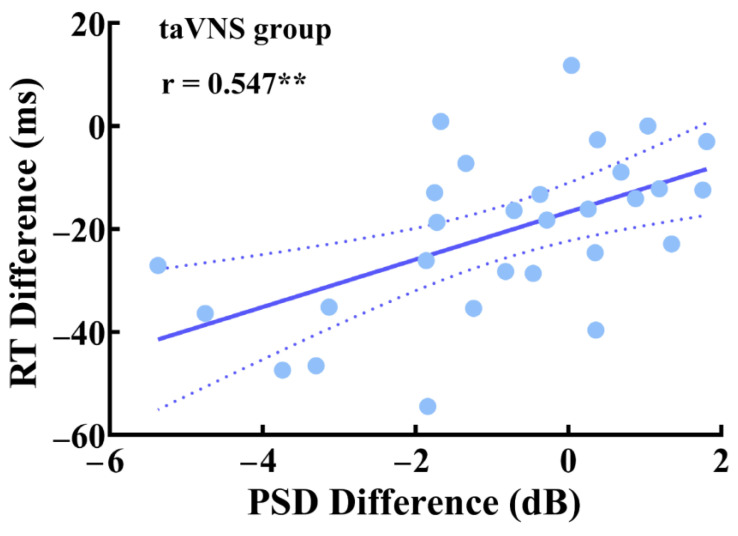
Partial correlation between the RT difference and the PSD difference of high-α oscillations in the taVNS group under the close-eye resting state. ** *p* < 0.01.

## Data Availability

The data and code that support the findings of this study are available from the corresponding author upon reasonable request.

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
