# Peer review of "Transcutaneous Auricular Vagus Nerve Stimulation Facilitates Cortical Arousal and Alertness"

_ijerph, 2023, doi:10.3390/ijerph20021402_

Round 1

Reviewer 1 Report

This study investigated the modulation effects of taVNS on central arousal. This is an interesting topic, the manuscript is well written. The following are more comments.

1.       Please provide rationale for using 25hz stimulation in this study.

2.       taVNS studies usually use the stimulation at other parts of the ear without vagus nerve distribution as control. In this study, the authors used first 30 second stimulation as control. One argument is the central arousal change may be derived from strong body sensory stimulation (not limited to vagus stimulation) as compared to no stimulation, the authors should include this point in the limitation section.

3.       In the group comparison, not sure why the authors perform a two-sample t-test, not the ANCOVA to control the age, gender and other covariates as they did in partial correlation analysis. It would be good to keep consistent in the data analysis.

4.       In the limitation section, the author may want to mention that the RT may not be significant after p value correction for two tests performed (one on left and another on right hands). If the authors have hypothesis on taVNS work on one hand but not on another hand, they should clarify this point in the manuscript.

Reviewer 2 Report

Manuscript ID: ijerph-2124347

Title: Transcutaneous auricular vagus nerve stimulation facilitates central

arousal and alertness

This looks like a manuscript with potential for publication, I just made a few suggestions.

1. Introduction

Lines 28-48

“Central arousal refers to the non-specific activation of the cerebral cortex associated with sleep-wake states [1]. It is controlled by the neuromodulatory system of the brain-29 stem nucleus [2], where the afferent branch of the vagus nerve projects to [3]. With a similar neural basis, central arousal is sometimes confused with alertness [1, 4]. From a psychological view, the alerting system is an essential part of attention and is in charge of maintaining optimal vigilance and performance [1, 5]. Thus, we preferred to define the term "central arousal" as the activation level of the cerebral cortex and "alertness" as the ability to maintain performance and keep vigilance in the present study. Both central arousal and alertness play crucial roles in our daily life. Exploring the modulatory effect of vagus nerve stimulation (VNS) on these two cores may provide a promising neuromodulation technique for improving drowsiness under heavy work in the healthy population. Furtherly, it will inspire therapies for treating patients with disorders of consciousness [6, 7] as well as attention deficit and hyperactivity disorder (ADHD, [8]).

It has been reported that VNS could lead to an enhancement of central (cortical) arousal [9, 10], as reflected by increased activation of locus coeruleus-norepinephrine (LC-NE, [11]), a large pupil dilation [12], and a reduction of alpha (α) oscillations in EEG [12, 13]. As an electrophysiological indicator of central arousal [14-16], α oscillations were demonstrated to be attenuated after 3.4 s transcutaneous auricular vagus nerve stimulation (taVNS) [12] and after two 120 s transcutaneous cervical vagus nerve stimulation (tcVNS) [13]. However, there was a lack of investigation on the modulatory effect of a long-bout continuous transcutaneous VNS on the central arousal under the resting state.”

A)    Review these first two paragraphs of the introduction, based on references 1, 2 and 4. I would suggest, above all, attention to the use of the word “central”. It seems to me to give a confusing idea as if only the cortex were the central nervous system.

B)    I suggest starting the introduction by commenting that there is no consensus among researchers in the search for an ideal term to describe states of cortical activation.

C)   In lines 36 and 37, isn't the replacement of the word “cores” by “conditions” more appropriate?

D)   In lines 41 and 48, I suggest removing the word “cortical”, as the sentence mentions the locus coeruleus, which is in the brainstem.

2. Materials and Methods

2.1. Subjects

Lines 70-71

“Sixty healthy right-handed subjects who were unacquainted with noninvasive neu-70 romodulation…”

E)    Wouldn't it be "neuromodulation"?

2.3. taVNS and control intervention

Lines 93-94

“In line with previous studies [11, 12, 24], stimulating electrodes were placed at cymba conchae which the ABVN mainly innervated ([3, 25, 26]; see Figure 1c).”

F)    In figure 1C, the electrodes are in the cimba and cava concha, unlike the description of the methodology, which mentions only the cimba concha.

Lines 93-105

G)   It was not clear in the text whether the stimulation was performed on the right, left or both sides.

2.4. Experimental Procedure

Lines 107-115

H)   It was not clear the use of anxiety and mainly depression instruments in a healthy population. Was it to demonstrate that the population was really healthy and/or if the behavioral test altered the baseline score? These results did not appear in the discussion.
